# ADAPTIVE DUAL-GRANULARITY PRUNING METHOD FOR LARGE LANGUAGE MODELS

## ABSTRACT

With the rapid development of large language models (LLMs), their parameter scales continue to expand, posing significant challenges for efficient deployment. Pruning, as a mainstream compression technique, can effectively reduce model size; however, it often suffers from robustness degradation and uncontrollable model size under high pruning ratios. In this work, we propose ADAP (**A**daptive **D**ual-Gr**a**nularity **P**runing) to address these two issues. ADAP ingeniously combines the global constraints of structured pruning with the flexibility of unstructured pruning, dynamically adjusting their respective proportions and introducing an intra-layer adaptive pruning ratio allocation mechanism, thereby overcoming the performance bottlenecks of conventional single-mode pruning. Moreover, we introduce *compression ratio* as a unified metric, replacing the commonly used *pruning ratio* to achieve precise control over model size. Experimental results demonstrate that ADAP significantly outperforms existing structured and unstructured pruning methods in high-compression scenarios, delivering better task performance while maintaining controllable model scale.

## 1 INTRODUCTION

In recent years, Large Language Models (LLMs) (Brown et al., 2020; OpenAI, 2024) have developed rapidly, demonstrating remarkable performance in tasks such as natural language understanding, generation, and reasoning. However, these models typically contain billions to trillions of parameters, and their massive scale results in substantial computational, inference, and storage costs, making the efficient utilization of model resources a pressing challenge.

To address this issue, model compression techniques have emerged, with pruning (Cun et al., 1990; Hassibi et al., 1993; Han et al., 2015) being a mainstream approach. Pruning can be broadly categorized into two types: structured pruning and unstructured pruning. **Structured pruning** (He & Xiao, 2024) removes redundant structural components (e.g., channels, layers, or neurons), directly reducing computation and memory usage. Yet, it damages the network structure, causing performance degradation at high pruning ratios and typically being limited to around 50% (Figure 1). By contrast, **unstructured pruning** (Han et al., 2015) eliminates individual weights with negligible contribution, achieving finer-grained compression and sustaining pruning ratios up to 70%. However, its reliance on sparse matrix formats requires storing additional indices, making *pruning ratio* an inaccurate indicator of model size. For example, in LLaMA-7B (Touvron et al., 2023), a pruning ratio of 45% is needed merely to reduce memory consumption.

Therefore, existing pruning techniques face two core challenges: **(1) how to ensure controllable model size, and (2) how to maintain stable performance at high pruning ratios.**

To address these challenges, we propose ADAP (**A**daptive **D**ual-Gr**a**nularity **P**runing), a framework that unifies structured and unstructured pruning to enhance model performance under high pruning ratios (Figure 1). ADAP consists of two key components: a dual-granularity pruning strategy and an intra-layer adaptive pruning ratio algorithm. First, we theoretically demonstrate the complementarity of structured and unstructured pruning, and for the first time, adaptively balance their proportions to more efficiently remove redundant weights. Second, we introduce the absolute value of cosine similarity as a redundancy metric, which better captures inter-layer information repetition than raw cosine similarity. Leveraging this metric, our algorithm extends adaptive pruning granularity from the inter-layer to the intra-layer level, enabling differentiated pruning ratios between attention and

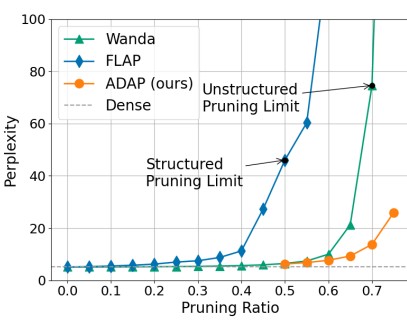 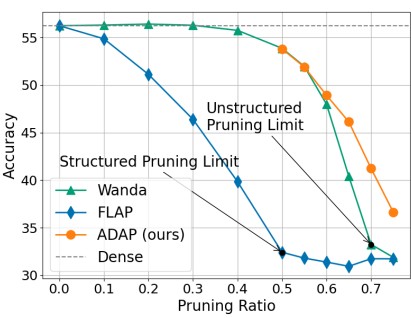

(a) Perplexity vs. Pruning Ratio    (b) Tasks Accuracy vs. Pruning Ratio

Figure 1: Perplexity (a) and Downstream Performance (b) of LLaMA2-7B under Structured (FLAP (An et al., 2023)), Unstructured (Wanda (Sun et al., 2024)) and ADAP Pruning Methods. Pruning limit marks highest pruning ratio before major performance drop. Perplexity ($\downarrow$) measures model confidence; Accuracy ($\uparrow$) reflects model performance on downstream tasks as Section 4 shown.

MLP sublayers. Finally, to overcome the limitations of using *pruning ratio* as a size metric in unstructured pruning, ADAP adopts *compression ratio*, ensuring controllable model size. Compared with prior approaches (Men et al., 2024; Dumitru et al., 2024; Chen et al., 2025), ADAP achieves superior pruning flexibility and model performance.

Our main contributions are listed as follows:

- We propose an adaptive dual-granularity pruning strategy, which for the first time integrates structured and unstructured pruning under theoretical guidance.
- We introduce the absolute value of cosine similarity as a new metric for assessing layer redundancy and propose a universal intra-layer adaptive pruning ratio algorithm.
- We adopt *compression ratio* instead of *pruning ratio* to measure model size, enabling compression-driven controllability of the model.

The rest of this paper is organized as follows. Section 2 reviews related work. Section 3 presents the design of ADAP, including the pruning strategy, intra-layer algorithm and size metric. Section 4 reports results, ablations and extended experiments. Section 5 discusses future directions and Section 6 concludes the paper.

## 2 RELATED WORK

LLMs have become central to natural language processing, but their inference and deployment incur high computational and memory costs. To address this, extensive research has focused on model compression (Kaplan et al., 2020), mainly categorized into four approaches: knowledge distillation (Hinton et al., 2015), low-rank decomposition (Tai et al., 2016), quantization (Dettmers et al., 2022; Frantar et al., 2023; Xiao et al., 2024), and pruning (He & Xiao, 2024; Han et al., 2015). Among these, pruning has gained attention for its structural simplicity and high compression efficiency, and is the primary focus of this study.

Pruning reduces model size by setting certain weights in LLMs to zero, and can be viewed as a form of extreme quantization. Based on granularity, pruning methods are categorized as unstructured or structured. Unstructured pruning removes individual weights and is fine-grained. Notable methods include SparseGPT (Frantar & Alistarh, 2023), which leverages layer-wise Hessian-guided pruning with dynamic mask updates; and Wanda (Sun et al., 2024), which evaluates weight importance based on both weight and activation magnitudes. Structured pruning removes weight groups in regular patterns, offering coarse-grained sparsity. Representative works include FLAP (An et al., 2023), which evaluates channel stability to guide pruning; and FASP (Hu et al., 2025), a structured extension of Wanda that assesses column importance via weighted sums of weights and activations, while incorporating cross-layer dependencies for error compensation.

Recent research introduces hierarchical redundancy into LLMs compression to further improve compression limits. ShortGPT (Men et al., 2024) analyzes layer-wise input-output cosine similarity to derive a metric for assessing layer importance, enabling entire-layer removal with minimal performance loss. DynamicSlicing (Dumitru et al., 2024) extends this idea by integrating dynamic pruning into SliceGPT, providing better adaptability than static approaches.

Despite their advantages, existing methods have limitations. Unstructured pruning requires storing sparse matrix indices, limiting compression efficiency; structured pruning is prone to abrupt performance drops and lacks robustness at high pruning ratios; hierarchical pruning, due to its coarse-grained nature, cannot fully exploit intra-layer redundancy. To overcome these issues, we propose an adaptive dual-granularity pruning strategy, which breaks through the bottlenecks of existing methods and improves both accuracy and stability under high pruning ratios.

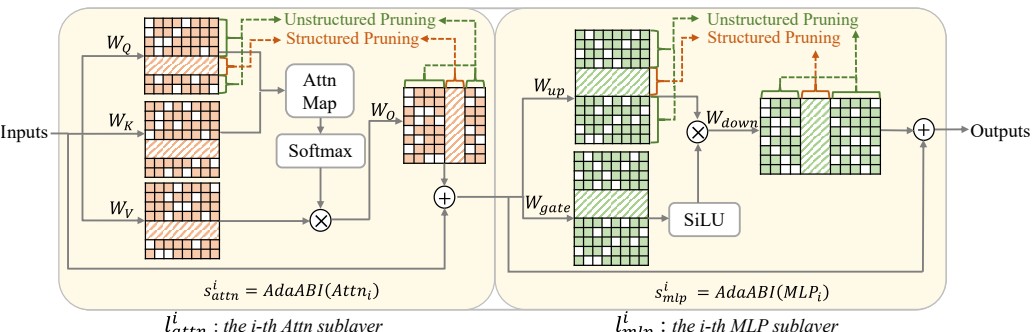

Figure 2: Architecture of ADAP on LLaMA. At $i$-th layer, $s^i_{\text{attn}}$ and $s^i_{\text{mlp}}$ are pruning ratios for attention and MLP sublayers. Our adaptive pruning algorithm (AdaABI) determines these ratios, then structured and unstructured pruning are applied simultaneously.

## 3 DESIGN OF ADAP

In this section, we first introduce ADAP, our proposed adaptive dual-granularity pruning strategy. ADAP consists of two key components: dual-granularity pruning and intra-layer adaptive pruning ratio algorithm. Next, we analyze the limitations of *pruning ratio* and use *compression ratio* for evaluating the compression effect of the model.

### 3.1 OVERVIEW OF ADAP

As shown in Figure 2, ADAP achieves adaptive dual-granularity pruning by first using the proposed absolute-value cosine similarity metric to quantify layer redundancy, providing a theoretical basis for pruning sensitivity. Based on this, an intra-layer adaptive pruning algorithm assigns fine-grained ratios to the attention and MLP sublayers, enabling precise intra-layer control. To balance the global pruning effect, structured and unstructured pruning are applied synchronously, mitigating the limitations of single pruning strategies.

### 3.2 DUAL-GRANULARITY PRUNING STRATEGY

Structured pruning and unstructured pruning exhibit inherent complementarity in terms of constraint and flexibility: the former ensures hardware efficiency and global sparsity consistency, while the latter removes redundant weights at a finer granularity. Motivated by this, we propose to jointly model both types of pruning to overcome the performance bottleneck of relying on single pruning strategies.

From the fundamental theorem of combinatorial optimization, for any objective function $f$ and two feasible domains $A \subseteq B$, we have:

$$\min_{x \in B} f(x) \leq \min_{x \in A} f(x). \tag{1}$$

In the pruning setting, feasible domains correspond to different sparsity patterns:

- $\mathcal{F}_{\text{channel}}$: feasible space of channel-level (structured) pruning, $s_c$ denotes its pruning ratio.

- $\mathcal{F}_{\text{weight}}$: feasible space of weight-level (unstructured) pruning, $s_w$ denotes its pruning ratio.

- $\mathcal{F}_{\text{joint}}$: feasible space of joint pruning, with $\mathcal{F}_{\text{channel}}, \mathcal{F}_{\text{weight}} \subseteq \mathcal{F}_{\text{joint}}$.

$$\mathcal{F}_{\text{channel}}(s_c) = \left\{ W \odot M_c \mid M_c \in M_{\text{channel}}(s_c), \ M_{\text{channel}} = \mathbf{m1}^\top, \ \mathbf{m} \in \{0,1\}^C \right\} \quad (2a)$$

$$\mathcal{F}_{\text{weight}}(s_w) = \left\{ W \odot M_w \mid M_c \in M_{\text{weight}}(s_w), \ M_{\text{weight}} \in \{0,1\}^{\text{shape}(W)} \right\} \quad (2b)$$

$$\mathcal{F}_{\text{joint}}(s_c, s_w) = \left\{ W \odot M_c \odot M_w \mid M_c \in M_{\text{channel}}(s_c), \ M_w \in M_{\text{weight}(s_w)} \right\} \quad (2c)$$

Thus, the optimal loss increment of joint pruning satisfies:

$$\Delta L_{\text{joint}} \le \Delta L_{\text{channel}}, \Delta L_{\text{joint}} \le \Delta L_{\text{weight}} \quad (3)$$

This indicates that joint optimization is theoretically guaranteed to achieve solutions no worse than either individual pruning strategy.

Formally, we define the pruned weights of layer $i$ as:

$$\hat{W}^{(i)} = M_{\text{struct}}^{(i)}(k_1, k_2) \odot M_{\text{unstr}}^{(i)}(k_1, k_2) \odot W^{(i)} \quad (4)$$

where $M_{\text{struct}}$ and $M_{\text{unstr}}$ denote the structured and unstructured pruning masks, respectively. The hyperparameter $k_1 \in [0,1]$ controls the proportion of structured pruning, and $k_2 \in [0,1]$ regulates the variation amplitude of pruning ratios.

Our objective is to minimize the task loss:

$$L_{\min} = \min_{k_1, k_2} L_{\text{task}}(W; k_1, k_2) \quad (5)$$

and we use perplexity (PPL) as an intuitive metric:

$$\text{PPL}_{\min} = \exp(L_{\min}) \quad (6)$$

To solve for the optimal hyperparameters $(k_1, k_2)$, we adopt a coarse-to-fine grid search to identify the best configuration at each preset pruning ratio. For each model family, we use a separate regression function to derive model-specific default hyperparameter settings. We do not use Bayesian optimization here, since the search space is only two-dimensional; grid search converges faster and provides more stable approximation of the global optimum in this context. We further provide an approximate polynomial fitting analysis of $(k_1, k_2)$ as a function of the pruning ratio $s$ in Appendix A.3, where we show that the fitted functions yield near-optimal pruning configurations across LLaMA family models.

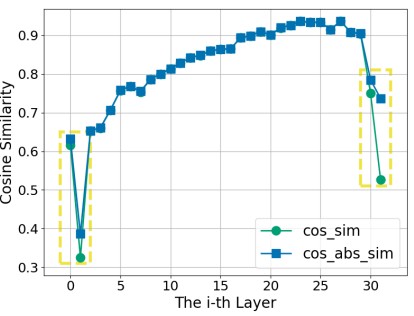
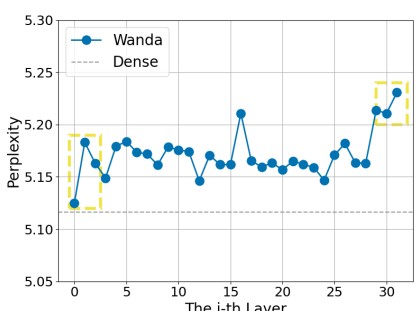

(a) Cosine Similarity and Absolute across Layers     (b) Perplexity across Layers (50% Wanda Pruning)

Figure 3: Cosine Similarity and Perplexity across Layers of LLaMA2-7B. Yellow boxes indicate layers with large differences between the two metrics.

### 3.3 Intra-layer Adaptive Pruning Algorithm

Existing works (Chen et al., 2025; Dumitru et al., 2024) commonly employ cosine similarity, denoted as $BI_i$ (Men et al., 2024) shown in Equation 7 to evaluate layer redundancy. However, we observe that cosine similarity fluctuates strongly in the first and last layers, while perplexity remains almost unchanged when pruning them (Figure 3), suggesting that cosine similarity does not faithfully reflect redundancy. The key issue is that redundancy corresponds to information substitutability rather than strict vector alignment: sparse inputs in early layers and aggregated features in later layers may flip directions without reducing correlation. Thus, we adopt the absolute cosine similarity, $ABI_i$ in Equation 8, which mitigates these misjudgments and provides more reliable guidance for adaptive pruning, ultimately leading to superior performance.

$$\text{BI}_i = E_{X,t} \frac{X_{i,t}^T X_{i+1,t}}{\|X_{i,t}\|_2 \|X_{i+1,t}\|_2} \tag{7}$$

$$\text{ABI}_i = E_{X,t} \left| \frac{X_{i,t}^T X_{i+1,t}}{\|X_{i,t}\|_2 \|X_{i+1,t}\|_2} \right| \tag{8}$$

The adaptive pruning algorithm $SLR$, as defined in Equation 9 and based on the original metric $BI$, demonstrates limited robustness when confronted with extreme values, potentially leading to pruning ratio overflow. To address this issue, we propose a novel adaptive algorithm $AdaABI$ shown in Equation 10, which introduces a hyperparameter $k_2$ to constrain the variation range of layer-wise pruning ratios, ensuring that the overall pruning ratio remains unchanged.

$$\text{SLR}(L_i) = \text{BI}_i \cdot \frac{s}{\overline{\text{BI}_i}} \tag{9}$$

$$\text{AdaABI}(L_i) = (\text{ABI}_i - \overline{\text{ABI}_i}) \cdot s \cdot k_2 + s \tag{10}$$

As demonstrated by Equations 11 and 12, $AdaABI$ guarantees a constant overall pruning ratio.

$$\sum_{i=1}^{n} \left( \text{ABI}_i - \overline{\text{ABI}_i} \right) = 0 \tag{11}$$

$$\sum_{i=1}^{n} \text{AdaABI}(L_i) = n \cdot s \tag{12}$$

Furthermore, motivated by the work (Zhong et al., 2022), we treat the attention and MLP sublayers as two independent modules. Our experiments show that $ABI$ values and post-pruning perplexity metrics differ significantly between the attention and MLP sublayers (see Figure 4). Consequently, we propose an intra-layer adaptive pruning ratio algorithm that allocates ratios per sublayer, offering finer granularity and superior performance while preserving the overall pruning ratio.

$$\text{AdaABI}(Attn_i) = (\text{ABI}_{\text{attn}_i} - \overline{\text{ABI}_{\text{attn}_i}}) \cdot s \cdot k_2 + s \tag{13}$$

$$\text{AdaABI}(MLP_i) = (\text{ABI}_{\text{mlp}_i} - \overline{\text{ABI}_{\text{mlp}_i}}) \cdot s \cdot k_2 + s \tag{14}$$

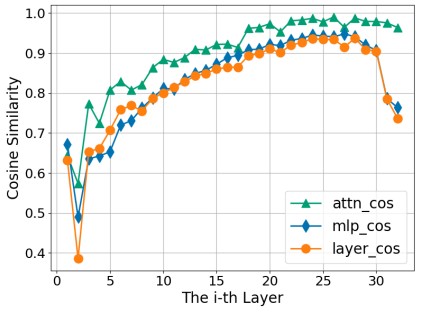
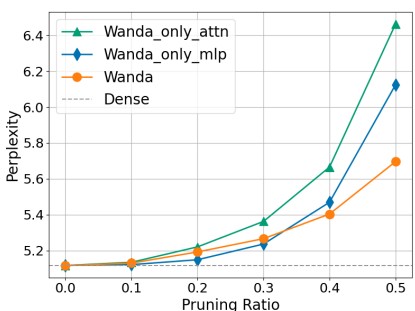

(a) ABI under Layer vs. Sublayer Pruning    (b) Perplexity under Layer vs. Sublayer Pruning

Figure 4: ABI and Perplexity of LLaMA2-7B under Layer and Sublayer Pruning.

### 3.4 Metric for Evaluating Model Size

The pruning ratio $s$ is the most commonly used metric for evaluating model compression methods. However, for unstructured pruning, due to its sparse matrix storage method, pruning ratio cannot reflect the actual reduction in model size. Therefore, we use the metric called *compression ratio* to reflect the actual model size after pruning, and together with the perplexity metric, to reflect the performance of the pruned model. In theory, the compression ratio can be estimated using the following formula:

$$c_{str} = \frac{bytes \cdot m \cdot n \cdot (1-s)}{m \cdot n \cdot bytes} \tag{15}$$

$$c_{unstr} = \frac{(bytes + \log_2 m + \log_2 n) \cdot m \cdot n \cdot (1-s)}{m \cdot n \cdot bytes} \tag{16}$$

$$c_{ours} = \frac{(bytes + \log_2 m + \log_2(n - n \cdot s \cdot k_1)) \cdot m \cdot n \cdot (1-s)}{m \cdot n \cdot bytes} \tag{17}$$

Equations 15, 16 and 17 represent the compression ratio for structured pruning, unstructured pruning and ADAP, respectively. Here, taking the $W_{up}$ as an example, $m$ and $n$ denote the input and output dimensions, $bytes$ is the number of bytes per element, $s$ represents the pruning ratio and $k_1$ indicates the proportion of structured pruning within ADAP.

As shown in Table 4 in Section 4, under the compression ratio metric, unstructured pruning performs worse than structured pruning due to the extra storage required for sparse matrix row and column indices. In contrast, ADAP leverages dual-granularity optimization, which both reduces the bitwidth of sparse matrix indices and efficiently removes redundant weights. Consequently, ADAP achieves superior compression ratios compared to either structured or unstructured pruning alone.

## 4 Experiments

In this section, we validate the performance of ADAP through comprehensive experiments. We begin by detailing our experimental settings, followed by a comparison of perplexity and zero-shot results for the pruned models obtained via ADAP and various baseline methods, as well as the results under the *compression ratio* metric.

**Models and baseline methods.** ADAP uses the structured metric FLAP (An et al., 2023) defined as $\frac{1}{N-1} \sum_{n=1}^{N} \|X_{n,j,:}^l - \bar{X}_{:,j,:}^l\|_2^2 \cdot \|W_{:,j}^l\|_2^2$ and the unstructured metric Wanda (Sun et al., 2024) defined as $|W_{ij}| \cdot \|X_j\|_2$. We compare ADAP with these two methods to evaluate its performance on the LLaMA (Touvron et al., 2023) family with sizes ranging from 7B to 65B downloaded from HuggingFace's Transformers library (Wolf et al., 2020).

**Datasets and benchmarks.** We evaluate the models' perplexity under various pruning ratios using 128 randomly sampled calibration examples from the WikiText2 (Merity et al., 2016) and C4 (Raffel et al., 2023) datasets, each with a sequence length of 2048. Additionally, we compare the zero-shot accuracy across the following four aspects: **Reasoning**: ARC-challenge, ARC-easy (Clark et al., 2018), HellasWag (Zellers et al., 2019), PIQA (Bisk et al., 2019), WindoGrande (Sakaguchi et al., 2019), MathQA (Amini et al., 2019). **Knowledge**: BoolQ (Clark et al., 2019),OpenbookQA (Mihaylov et al., 2018). **Examination**: CMMLU (Li et al., 2024). **Understanding**: Race (Lai et al., 2017).

### 4.1 Main Results

**Perplexity and zero-shot results**. We present the perplexity results for the pruned LLaMA models on WikiText in Figure 5 and Table 2. The results for LLaMA2, as well as for LLaMA on C4, are provided in Appendix A.1. Data and figures consistently demonstrate that ADAP performs significantly better than baseline methods at a high pruning ratio, breaking through the pruning ratio limits of structured and unstructured pruning and maintaining stability of the models.

Additionally, the zero-shot results for the pruned LLaMA-7B and LLaMA2-7B models are presented in Table 3, where ADAP outperforms the baseline methods by 2.55% and 11.56% at pruning ratios of 60% and 70% for LLaMA-7B, and by 1.96% and 24.04% at pruning ratios of 60% and 70% for

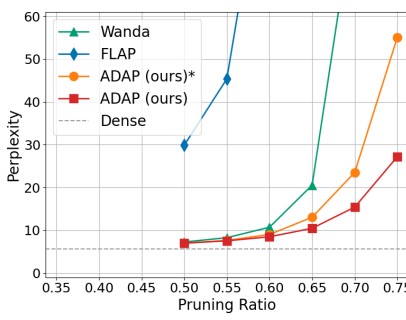 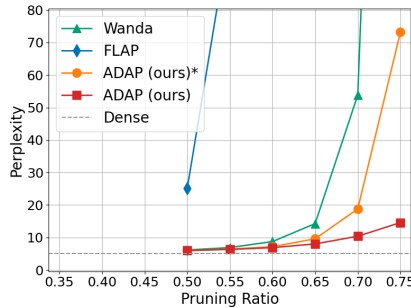

(a) Perplexity vs. Pruning Ratio for LLaMA-7B  (b) Perplexity vs. Pruning Ratio for LLaMA-13B

Figure 5: Perplexity ($\downarrow$) for LLaMA-7B (a) and LLaMA-13B (b) on WikiText. ADAP* represents pruned models without the adaptive pruning ratio algorithm.

LLaMA2-7B. The downstream task performance of the 13B LLaMA models under different pruning ratios is presented in the Appendix A.1. Therefore, the accuracy of the models is significantly improved under a high pruning ratio.

**The results under the metric *compression ratio***. Compression ratios of the pruning methods are computed using Equations 15, 16 and 17. We present the perplexity results of LLaMA models of different sizes under the *compression ratio* metric in Table 4 and Appendix A.1. ADAP performs significantly better than the baseline methods at high compression ratios.

**Pruning time.** We evaluated the pruning time of the methods to assess their computational efficiency. As shown in Table 1, although ADAP incurs slightly higher overhead due to dual-granularity optimization, the additional runtime is minimal compared to the baseline, indicating that our approach introduces negligible computational cost while achieving superior pruning performance.

| Method | Pruning Time |
|---|---|
| Wanda | 78s |
| FLAP | 295s |
| ADAP | 302s |

Table 1: Comparison of LLaMA-7B Pruning Times.

| Method | Pruning Ratio | LLaMA | | | |
|---|---|---|---|---|---|
| | | 7B | 13B | 30B | 65B |
| Dense | 0% | 5.67 | 5.09 | 4.10 | 3.53 |
| Wanda | 50% | 7.26 | 6.15 | 5.25 | 4.55 |
| FLAP | 50% | 29.86 | 25.07 | 90.64 | 817.03 |
| ADAP (ours)* | 50% | **6.98** | **5.99** | 5.25 | 4.50 |
| ADAP (ours) | 50% | 6.99 | **5.99** | **5.08** | **4.49** |
| Wanda | 60% | 10.71 | 8.75 | 6.56 | 5.67 |
| FLAP | 60% | 105.26 | 576.08 | 683.87 | 15311.95 |
| ADAP (ours)* | 60% | 9.02 | 7.17 | 5.93 | 5.27 |
| ADAP (ours) | 60% | **8.47** | **6.88** | **5.89** | **5.14** |
| Wanda | 70% | 86.18 | 53.85 | 17.48 | 12.25 |
| FLAP | 70% | 481.84 | 60957.43 | 1570.72 | 5745.54 |
| ADAP (ours)* | 70% | 23.53 | 18.79 | 9.73 | 7.52 |
| ADAP (ours) | 70% | **15.41** | **10.39** | **7.56** | **6.18** |

Table 2: Perplexity ($\downarrow$) of Pruned LLaMA Models on WikiText under Different Pruning Ratios. ADAP* represents pruned models without the adaptive pruning ratio algorithm.

## 4.2 ABLATION STUDIES

**The impact of hybrid pruning methodology**. In this experiment, we set the hyperparameter $k_2$ to zero and compared it with the baseline methods after ignoring the effect of the adaptive algorithm. As shown in Figure 5 and Table 2, the hybrid pruning methodology (ADAP*) can achieve better results than the baseline methods.

| Model | Method | Pruning Ratio | ARC-c | ARC-e | Hellaswag | PIQA | WinoGrande | MathQA | BoolQ | OBQA | CMMLU | Race | Mean |
|---|---|---|---|---|---|---|---|---|---|---|---|---|---|
| | Dense | 0% | 44.80 | 72.85 | 76.18 | 79.16 | 70.01 | 26.53 | 75.11 | 44.40 | 26.24 | 40.19 | 55.55 |
| | Wanda | 60% | 33.02 | 56.90 | 58.70 | **73.01** | 63.06 | 22.85 | **68.99** | 36.40 | 25.27 | **35.79** | 47.40 |
| | FLAP | 60% | 25.26 | 27.69 | 27.16 | 51.36 | 50.43 | 19.70 | 41.99 | 27.20 | **25.57** | 22.78 | 31.91 |
| LLaMA | ADAP (ours) | 60% | **35.32** | **59.18** | **65.25** | 72.85 | **63.69** | **23.66** | 66.88 | **38.60** | 25.23 | 35.41 | **48.61** |
| | Wanda | 70% | 20.99 | 32.87 | 31.13 | 55.71 | 51.07 | **22.57** | 58.50 | 26.60 | 24.76 | 26.12 | 35.03 |
| | FLAP | 70% | **29.69** | 25.63 | 25.84 | 49.08 | 52.09 | 22.29 | 47.19 | 26.60 | **24.98** | 23.25 | 32.66 |
| | ADAP (ours) | 70% | 27.82 | **33.29** | **41.84** | **58.54** | **56.35** | 20.17 | **63.79** | **35.00** | 24.87 | **29.09** | **39.08** |
| | Dense | 0% | 46.25 | 74.54 | 76.01 | 79.11 | 68.90 | 28.64 | 77.68 | 44.20 | 27.44 | 39.62 | 56.24 |
| | Wanda | 60% | 32.76 | **60.40** | 58.48 | 71.82 | 65.51 | **24.83** | 66.82 | 38.60 | 25.37 | 35.22 | 47.98 |
| | FLAP | 60% | 27.22 | 26.98 | 25.77 | 51.52 | 49.09 | 19.70 | 41.44 | 23.80 | 24.83 | 23.44 | 31.38 |
| LLaMA2 | ADAP (ours) | 60% | **34.98** | 58.00 | **63.41** | **72.58** | **65.59** | 23.93 | **67.52** | **39.60** | 25.37 | **38.18** | **48.92** |
| | Wanda | 70% | 22.18 | 30.26 | 29.87 | 53.37 | 50.83 | **22.37** | 46.79 | 26.20 | **25.34** | 25.17 | 33.24 |
| | FLAP | 70% | 26.88 | 26.01 | 25.86 | 52.45 | 47.75 | 20.14 | 43.82 | 26.80 | 25.33 | 22.30 | 31.73 |
| | ADAP (ours) | 70% | **29.69** | **47.43** | **44.40** | **65.61** | **54.30** | 22.32 | **62.14** | **31.20** | 25.32 | **29.86** | **41.23** |

Table 3: Zero-shot Accuracy (↑) of Pruned LLaMA-7B and LLaMA2-7B Models under Different Pruning Ratios.

| Method | Compression Ratio | Pruning Ratio | Perplexity |
|---|---|---|---|
| Dense | 100.00% | 0.00% | 5.67 |
| Wanda | 60.00% | 66.08% | 22.24 |
| FLAP | 60.00% | 40.00% | 12.53 |
| ADAP (ours) | 60.00% | 65.74% | **11.42** |
| Wanda | 50.00% | 71.74% | 139.38 |
| FLAP | 50.00% | 50.00% | 28.86 |
| ADAP (ours) | 50.00% | 71.36% | **18.47** |
| Wanda | 40.00% | 77.39% | 1265.01 |
| FLAP | 40.00% | 60.00% | 105.26 |
| ADAP (ours) | 40.00% | 76.99% | **44.28** |

Table 4: Perplexity (↓) of LLaMA-7B Model on WikiText under Different Compression Ratios.

**The impact of adaptive pruning ratio algorithm**. We conducted comparative experiments on the baseline method FLAP (An et al., 2023) using the LLaMA2-7B model. For completeness, we also compared against FASP (Hu et al., 2025) and the results are provided in Appendix A.2. (1) We used the DynamicSlicing adaptive algorithm (Dumitru et al., 2024) to

| Method | 10% | 20% | 30% | 40% | 50% |
|---|---|---|---|---|---|
| FLAP$_{BI}$ | **5.49** | 6.18 | 7.34 | 9.53 | 14.10 |
| FLAP$_{ABI}$ | **5.49** | **6.13** | **7.32** | **9.35** | **13.77** |

Table 5: Perplexity (↓) under Different Pruning Ratios for FLAP on LLaMA2-7B by Two Metrics.

compare the cosine similarity metric ($BI$) (Men et al., 2024) and the absolute cosine similarity metric ($ABI$), as shown in Table 5. $ABI$ achieved better results than $BI$ on both baseline methods. (2) As shown in Table 6, the adaptive algorithm ($AdaABI$) proposed in this paper consistently outperforms the DynamicSlicing adaptive algorithm ($SLR$) across both baseline methods. (3) Comparing the effects of our adaptive algorithm on inter layer and intra layer interactions, as shown in Table 6, the intra layer adaptive effect is better than the inter layer effect on both baseline methods.

| Method | 10% | 20% | 30% | 40% | 50% |
|---|---|---|---|---|---|
| FLAP | 5.50 | 6.23 | 7.52 | 11.27 | 46.04 |
| FLAP_SLR_inter-layer | 5.49 | 6.18 | 7.34 | 9.53 | 14.10 |
| FLAP_AdaABI_inter-layer | **5.48** | **6.15** | 7.31 | 9.24 | 13.77 |
| FLAP_AdaABI_intra-layer | **5.48** | **6.15** | **7.23** | **9.14** | **13.76** |

Table 6: Perplexity (↓) under Different Pruning Ratios for FLAP on LLaMA2-7B under Dynamic-Slicing and Our Adaptive Algorithms.

## 4.3 EXTENDED EXPERIMENTS

We evaluated the performance of ADAP on the OPT and DeepSeek models under different structured and unstructured pruning metrics.

**OPT models.** We replace the structured pruning metric with FASP (Hu et al., 2025) defined as $\sum |W_{ij}| \cdot \left\| X_{(:,j)} \right\|_2$ and the unstructured pruning metric with SparseGPT (Frantar & Alistarh,

2023) formulated as $|W|^2/\text{diag}((XX^\top + \lambda I)^{-1})_{ij}$. We conducted experiments on the OPT models (Zhang et al., 2022) and compared them with baseline methods, including FASP and SparseGPT. We present the perplexity results of OPT models of different sizes on WikiText in Table 7. ADGP performs better than the baseline method at high pruning ratios.

**Deepseek models.** To further validate the generality of ADAP, we apply it to DeepSeek models, using the structured metric FASP and unstructured pruning metric Wanda. Experiments were conducted on DeepSeek-V2 (DeepSeek-AI et al., 2024), and the results were compared with the corresponding baseline pruning methods. As shown in Table 8, ADAP consistently outperforms the baseline methods, especially at high pruning ratios, demonstrating its effectiveness across mixture-of-experts (MoE) architectures as well.

Extended experiments demonstrate the generality of ADAP, showing that it can be effectively applied across different models and compatible with any structured or unstructured pruning metrics, while maintaining strong performance even at high pruning ratios.

| Method | Pruning Ratio | OPT-125M | OPT-1.3B |
|---|---|---|---|
| Dense | 0% | 27.65 | 14.63 |
| SparseGPT | 50% | 33.20 | 26.87 |
| FASP | 50% | 102.95 | 78.03 |
| ADAP (ours) | 50% | **32.88** | **25.29** |
| SparseGPT | 60% | 46.25 | 31.29 |
| FASP | 60% | 189.06 | 170.81 |
| ADAP (ours) | 60% | **42.49** | **28.29** |

Table 7: Perplexity ($\downarrow$) of Pruned OPT Model on WikiText under Different Pruning Ratios.

| Method | Pruning Ratio | WikiText | C4 |
|---|---|---|---|
| Dense | 0% | 6.31 | 9.325 |
| Wanda | 60% | 12.33 | 17.64 |
| FASP | 60% | 34.19 | 40.73 |
| ADAP (ours) | 60% | **10.58** | **15.01** |

Table 8: Perplexity ($\downarrow$) of Pruned Deepseek-V2 Model on WikiText under 60% Pruning Ratio.

## 5 FUTURE WORK

In future work, ADAP can be extended to integrate structured pruning with semi-structured methods (e.g., 2:4, 4:8) (Mishra et al., 2021). While semi-structured pruning reduces memory usage compared to unstructured pruning at $50\%$ pruning ratio, its ratio is fixed and cannot be adjusted. By combining it with structured pruning, we can enable flexible pruning ratio while leveraging the efficiency of semi-structured patterns, thereby improving accuracy and deployment on edge devices under given compression budgets.

## 6 CONCLUSION

In this work, we propose ADAP (**A**daptive **D**ual-Gr**a**nularity **P**runing) to address the challenges of robustness degradation and uncontrollable model size in pruning large language models under high compression settings. By unifying the global regularity of structured pruning and the flexibility of unstructured pruning, ADAP dynamically balances their contributions and introduces an intra-layer adaptive pruning mechanism to better capture redundancy across different sublayers. Furthermore, we replace the conventional pruning ratio with the compression ratio as a more precise metric for evaluating the true size of compressed models. Extensive experiments on Dense and MoE models validate that ADAP consistently outperforms baseline methods, achieving superior task performance and stability while maintaining controllable model scale even at high compression ratios.

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

## A ADDITIONAL EXPERIMENTS

**Models and baseline methods.** We compare ADAP with structured and unstructured pruning methods, including FLAP (An et al., 2023) and Wanda (Sun et al., 2024), in the LLaMA (Touvron et al., 2023) families downloaded from HuggingFace's Transformers library (Wolf et al., 2020).

**Datasets and benchmarks.** We evaluate the models' perplexity under various pruning ratios using 128 randomly sampled calibration examples from the WikiText2 (Merity et al., 2016) and C4 (Raffel et al., 2023) datasets, each with a sequence length of 2048. Additionally, we compare the zero-shot accuracy across the following four aspects: **Reasoning**: ARC-challenge, ARC-easy (Clark et al., 2018), HellasWag (Zellers et al., 2019), PIQA (Bisk et al., 2019), WindoGrande (Sakaguchi et al., 2019), MathQA (Amini et al., 2019). **Knowledge**: BoolQ (Clark et al., 2019),OpenbookQA (Mihaylov et al., 2018). **Examination**: CMMLU (Li et al., 2024). **Understanding**: Race (Lai et al., 2017).

| Method | Pruning Ratio | LLaMA2 | |
|---|---|---|---|
| | | 7B | 13B |
| Dense | 0% | 5.11 | 4.57 |
| Wanda | 50% | 6.46 | 5.58 |
| FLAP | 50% | 46.04 | 95.81 |
| ADAP (ours)* | 50% | 6.27 | 5.44 |
| ADAP (ours) | 50% | **6.26** | **5.42** |
| Wanda | 60% | 10.02 | 7.93 |
| FLAP | 60% | 124.98 | 2586.31 |
| ADAP (ours)* | 60% | 8.04 | 6.65 |
| ADAP (ours) | 60% | **7.71** | **6.47** |
| Wanda | 70% | 74.58 | 44.38 |
| FLAP | 70% | 830.79 | 13684.27 |
| ADAP (ours)* | 70% | 17.87 | 21.36 |
| ADAP (ours) | 70% | **13.77** | **11.34** |

Table 9: Perplexity (↓) of Pruned LLaMA2 Models on WikiText under Different Pruning Ratios. ADAP* represents pruned models without the adaptive pruning ratio algorithm.

### A.1 ADDITIONAL RESULTS FOR MAIN EXPERIMENTS

**Perplexity and zero-shot results**. Perplexity results are reported for pruned LLaMA2 models on WikiText (Merity et al., 2016) in Table 9, and for LLaMA models on C4 in Table 10. Data shows that ADAP performs significantly better than baseline methods at a high pruning ratio. The zero-shot results for the pruned LLaMA-13B and LLaMA2-13B models are presented in Table 11. ADAP outperforms the baseline methods by 1.95% and 24.93% at pruning ratio of 60% and 70% for LLaMA-13B, and by 1.61% and 26.47% at pruning ratio of 60% and 70% for LLaMA2-13B. Therefore, the accuracy of the models is significantly improved under a high pruning ratio.

**The results under the metric *compression ratio*.** Compression ratios of the pruning methods are computed using Equations 15, 16 and 17. We present the perplexity results of LLaMA-13B model under the *compression ratio* metric in Table 12. ADAP performs significantly better than the baseline methods at high compression ratios.

| Method | Pruning Ratio | LLaMA | | |
|---|---|---|---|---|
| | | 7B | 13B | 30B |
| Dense | 0% | 7.34 | 6.80 | 6.13 |
| Wanda | 60% | 14.34 | 11.73 | 9.50 |
| FLAP | 60% | 271.53 | 539.97 | 903.32 |
| ADAP (ours) | 60% | **12.90** | **10.91** | **9.01** |

Table 10: Perplexity (↓) of Pruned LLaMA Models on C4 under 60% Pruning Ratio.

| Model | Method | Pruning Ratio | ARC-c | ARC-e | Hellaswag | PIQA | WinoGrande | MathQA | BoolQ | OBQA | CMMLU | Race | Mean |
|---|---|---|---|---|---|---|---|---|---|---|---|---|---|
| | Dense | 0% | 47.70 | 74.75 | 79.06 | 80.09 | 72.93 | 29.65 | 77.89 | 44.80 | 26.21 | 39.71 | 57.28 |
| | Wanda | 60% | 36.43 | **64.44** | 67.02 | **75.24** | **69.30** | 25.61 | 71.35 | 42.40 | **25.61** | 39.33 | 51.67 |
| | FLAP | 60% | 26.71 | 28.83 | 29.23 | 51.74 | 50.28 | 20.45 | 39.02 | 24.60 | 24.68 | 23.73 | 31.93 |
| LLaMA | ADAP (ours) | 60% | **39.93** | 64.14 | **70.75** | 74.92 | 69.22 | **26.94** | **73.27** | **42.60** | 25.09 | **39.90** | **52.68** |
| | Wanda | 70% | 22.10 | 37.04 | 34.56 | 61.10 | 53.12 | 23.18 | 61.65 | 30.20 | **25.26** | 27.66 | 37.59 |
| | FLAP | 70% | 28.16 | 28.32 | 25.73 | 49.89 | 49.17 | 20.79 | 39.48 | 26.00 | 25.26 | 23.16 | 31.60 |
| | ADAP (ours) | 70% | **32.34** | **55.81** | **59.42** | **71.16** | **65.90** | **23.29** | **62.84** | **37.60** | 25.17 | **36.08** | **46.96** |
| | Dense | 0% | 49.06 | 77.44 | 79.37 | 80.52 | 72.14 | 32.04 | 80.55 | 45.20 | 34.77 | 40.48 | 59.16 |
| | Wanda | 60% | 40.02 | 64.52 | 65.67 | 75.84 | 68.51 | 26.86 | **77.09** | 39.80 | 25.97 | 38.66 | 52.29 |
| | FLAP | 60% | 25.51 | 27.78 | 28.88 | 53.10 | 49.80 | 19.95 | 38.32 | 27.20 | 25.27 | 24.02 | 31.98 |
| LLaMA2 | ADAP (ours) | 60% | **40.44** | **67.89** | **69.88** | **76.22** | **69.30** | **27.05** | 74.07 | **41.20** | **26.48** | **38.76** | **53.13** |
| | Wanda | 70% | 20.90 | 33.67 | 30.92 | 56.96 | 50.91 | 23.27 | 62.29 | 26.80 | 25.30 | 27.46 | 35.85 |
| | FLAP | 70% | 30.97 | 27.02 | 26.37 | 49.18 | 49.49 | 18.92 | 38.29 | 25.60 | 24.81 | 22.68 | 31.33 |
| | ADAP (ours) | 70% | **31.66** | **53.28** | **53.15** | **69.21** | **64.56** | **24.41** | **62.57** | **35.80** | 25.32 | **34.40** | **45.34** |

Table 11: Zero-shot Accuracy (↑) of Pruned LLaMA-13B and LLaMA2-13B Models under Different Pruning Ratios.

| Method | Compression ratio | Pruning Ratio | Perplexity |
|---|---|---|---|
| Dense | 100.00% | 0.00% | 5.09 |
| Wanda | 60.00% | 66.47% | 15.83 |
| FLAP | 60.00% | 40.00% | 14.28 |
| ADAP (ours) | 60.00% | 66.33% | **8.76** |
| Wanda | 50.00% | 72.06% | 81.28 |
| FLAP | 50.00% | 50.00% | 25.07 |
| ADAP (ours) | 50.00% | 71.92% | **13.64** |
| Wanda | 40.00% | 77.65% | 1541.27 |
| FLAP | 40.00% | 60.00% | 576.08 |
| ADAP (ours) | 40.00% | 77.25% | **19.87** |

Table 12: Perplexity (↓) of Pruned LLaMA-13B Model on WikiText under Different Compression Ratios.

## A.2 SUPPLEMENTARY ABLATION STUDIES

For completeness, we also report results on the FASP baseline (Hu et al., 2025). Consistent with the findings in the main paper, the proposed absolute cosine similarity metric ($ABI$) outperforms the cosine similarity metric ($BI$), and the adaptive algorithm ($AdaABI$) further surpasses the DynamicSlicing algorithm ($SLR$). As shown in Tables 13 and 14, these results verify that our conclusions hold across both FLAP and FASP, thereby demonstrating the robustness and generality of our approach.

| Method | 10% | 20% | 30% | 40% | 50% |
|---|---|---|---|---|---|
| FASP$_{\text{BI}}$ | **5.83** | **6.51** | 7.71 | 9.85 | 14.57 |
| FASP$_{\text{ABI}}$ | **5.83** | 6.52 | **7.65** | **9.66** | **14.07** |

Table 13: Perplexity ($\downarrow$) under Different Pruning Ratios for FASP on LLaMA2-7B by Two Metrics.

| Method | 10% | 20% | 30% | 40% | 50% |
|---|---|---|---|---|---|
| FASP | **5.80** | 6.55 | 7.90 | 10.85 | 16.78 |
| FASP_SLR_inter-layer | 5.83 | 6.51 | 7.71 | 9.85 | 14.57 |
| FASP_AdaABI_inter-layer | 5.83 | **6.49** | 7.65 | 9.72 | 14.08 |
| FASP_AdaABI_intra-layer | 5.83 | **6.49** | **7.63** | **9.56** | **13.62** |

Table 14: Perplexity ($\downarrow$) under Different Pruning Ratios for FASP on LLaMA2-7B under Dynamic-Slicing and Our Adaptive Algorithms.

### A.3 APPROXIMATE FITTING RESULTS FOR LLAMA

As shown in Table 15, the distributions of $k_1$ and $k_2$ under the same pruning ratio $s$ are highly consistent across LLaMA family models, which suggests that $s$ can serve as a unified variable for approximation. To capture this trend, we employ polynomial regression and report the fitting functions.

| $s$ | LLaMA | | | | LLaMA2 | | $s$ | LLaMA | | | | LLaMA2 | |
|---|---|---|---|---|---|---|---|---|---|---|---|---|---|
| | 7B | 13B | 30B | 65B | 7B | 13B | | 7B | 13B | 30B | 65B | 7B | 13B |
| 50% | 0.15 | 0.20 | 0.12 | 0.10 | 0.13 | 0.08 | 50% | 0.06 | 0.41 | 0.10 | 0.20 | 0.11 | 0.13 |
| 55% | 0.19 | 0.28 | – | – | 0.15 | 0.11 | 55% | 0.21 | 0.18 | – | – | 0.23 | 0.13 |
| 60% | 0.26 | 0.25 | 0.26 | 0.30 | 0.26 | 0.15 | 60% | 0.28 | 0.26 | 0.35 | 0.40 | 0.25 | 0.24 |
| 65% | 0.24 | 0.22 | – | – | 0.35 | 0.15 | 65% | 0.42 | 0.38 | – | – | 0.35 | 0.29 |
| 70% | 0.56 | 0.24 | 0.45 | 0.40 | 0.46 | 0.13 | 70% | 0.37 | 0.46 | 0.47 | 0.50 | 0.45 | 0.39 |
| 75% | 0.64 | 0.62 | – | – | 0.53 | 0.45 | 75% | 0.50 | 0.75 | – | – | 0.32 | 0.50 |

(a) $k_1$ vs. $s$ for different models          (b) $k_2$ vs. $s$ for different models

Table 15: Comparison of $k_1$ (a) and $k_2$ (b) across different $s$ values for LLaMA family models.

For $k_1$, linear, quadratic, and cubic fittings are evaluated, with the cubic polynomial achieving the best fit (Eq. 18, $R^2 = 0.9886$), where $R^2$ (coefficient of determination) measures how well the fitted function explains the variance of the observed data. A higher $R^2$ value indicates a better fit.

$$k_1(s) = 62.2840s^3 - 110.0146s^2 + 65.2948s - 12.8014, \quad R^2 = 0.9886 \tag{18}$$

For comparison, the linear and quadratic fittings are given by

$$k_1(s) = 1.5519s - 0.6812, \quad R^2 = 0.8682 \tag{19}$$

$$k_1(s) = 6.7679s^2 - 6.9079s + 1.9132, \quad R^2 = 0.9562 \tag{20}$$

For $k_2$, the fitting is sufficiently captured by a linear approximation:

$$k_2(s) = 1.4667s - 0.5883, \quad R^2 = 0.9835 \tag{21}$$

while the quadratic form provides a marginal improvement:

$$k_2(s) = 1.2500s^2 - 0.0958s - 0.1092, \quad R^2 = 0.9873 \tag{22}$$

In summary, we adopt the cubic polynomial (Eq. 18) to represent $k_1$ and the linear function (Eq. 21) to represent $k_2$, striking a balance between accuracy and simplicity in approximation.

We further validate the fitted functions on LLaMA-7B and LLaMA-13B (Table 16). The approximated $k_1$ and $k_2$ values consistently yield pruning performance very close to the optimal ADAP settings (e.g., 7.00 vs. 6.99 on LLaMA-7B at $50\%$ pruning ratio, 6.97 vs. 6.88 on LLaMA-13B at $60\%$ pruning ratio). In contrast, baseline methods such as Wanda and FLAP exhibit significantly worse performance, especially at higher pruning ratios. This demonstrates that the fitting approach not only provides a reliable approximation for hyperparameter selection, but also ensures near-optimal pruning configurations across LLaMA family models.

| Method | Pruning Ratio | LLaMA-7B | LLaMA-13B |
|---|---|---|---|
| Dense | 0% | 5.67 | 5.09 |
| Wanda | 50% | 7.26 | 6.15 |
| FLAP | 50% | 29.86 | 25.07 |
| ADAP (approximated) | 50% | 7.00 | **5.99** |
| ADAP | 50% | **6.99** | **5.99** |
| Wanda | 60% | 10.71 | 8.75 |
| FLAP | 60% | 105.26 | 576.08 |
| ADAP (approximated) | 60% | 8.72 | 6.97 |
| ADAP | 60% | **8.47** | **6.88** |

Table 16: Validation of fitted $k_1$ and $k_2$ on LLaMA-7B and LLaMA-13B.

