# OpenReview forum: "Adaptive Dual-Granularity Pruning Method for Large Language Models"
_ICLR.cc/2026/Conference — Submitted to ICLR 2026_

### Official Review · Reviewer_3CZk · 2025-10-25

**Soundness:** 2
**Presentation:** 2
**Contribution:** 2
**Rating:** 2
**Confidence:** 4

**Summary:**

This paper introduces ADAP (Adaptive Dual-Granularity Pruning), a novel framework designed to enhance the efficiency and controllability of pruning large language models (LLMs). The work addresses two long-standing issues in model compression: the instability of structured pruning at high pruning ratios and the lack of controllable model size in unstructured pruning. ADAP unifies both approaches into a single, theoretically grounded framework that dynamically balances structured and unstructured pruning based on model redundancy. The method leverages the complementarity between structured pruning, which ensures global consistency and hardware efficiency, and unstructured pruning, which provides fine-grained flexibility to remove redundant parameters. To further improve adaptability, the authors introduce an intra-layer adaptive pruning algorithm (AdaABI), which uses the absolute cosine similarity (ABI) as a more reliable redundancy measure compared to the traditional cosine similarity. This algorithm adaptively allocates pruning ratios within each Transformer layer, distinguishing between attention and MLP sublayers while maintaining a fixed global pruning level. Another key contribution (claimed by the authors) is the introduction of a compression ratio metric that more accurately reflects the effective reduction in model size, overcoming the limitations of the conventional pruning ratio metric that ignores the storage overhead of sparse matrices.

**Strengths:**

1. Novel unified pruning framework: The paper presents a theoretically grounded integration of structured and unstructured pruning into a single adaptive framework. This dual-granularity approach is both conceptually elegant and practically effective, addressing the weaknesses of each method while leveraging their complementary advantages.

2. Adaptive intra-layer pruning mechanism: The proposed AdaABI algorithm introduces an adaptive mechanism that dynamically allocates pruning ratios across attention and MLP sublayers. This fine-grained control improves pruning precision and preserves model robustness even at high compression levels.

3. The overall writing is easy to follow.

**Weaknesses:**

1. The searched sparsity allocation is not clear.  With the overall sparsity target, this paper uses the grid search to obtain the sparsity allocation for unstructured and structured pruning. However, the exact searched results (and the exact ratio setup for $k1$ and $k2$) are not shown in the paper. Currently, structured pruning can bring direct speedup but do harm to the performance, while the unstructured pruning needs the specified hardware design for speedup. Therefore, it is important to know the exact results for this to see if the following results comparsion is fair enough or not.

2. Some critical baselines are missing. The main contribution of this paper is the introduction of adaptive layer sparsity allocation. However, some previous works[1, 2] have explored this. The paper should add and compare these works in a fair manner.

3. The motivation of proposing the new metric. In fact, I don't see the necessity of proposing such compression ratio metric, which is actually very similar to the pruning ratio metric. The authors should state the key motivation of using this metric instead of the previous, widely-used one.




[1] Yin L, Wu Y, Zhang Z, et al. Outlier weighed layerwise sparsity (owl): A missing secret sauce for pruning llms to high sparsity[J]. arXiv preprint arXiv:2310.05175, 2023.

[2]Tang S, Sieberling O, Kurtic E, et al. Darwinlm: Evolutionary structured pruning of large language models[J]. arXiv preprint arXiv:2502.07780, 2025.

**Questions:**

1.  Some formulations of this paper are not clear. I'd suggest the authors check their formula carefully. For example, what is $\overline{ABI_i} $ in Equation 10? I would guess it is the average of $ABI$. The author should revise this and make it clearer.

---

> ### Author Response · Authors · 2025-12-03
> **Rebuttal to Reviewer 3CZk**
>
> Dear Reviewer,
>
> Thank you very much for your thorough review and valuable comments. We respectfully address the concerns raised below:
>
> First, regarding the pruning ratio allocation between structured and unstructured pruning, we would like to clarify that the search process for the hyperparameters $k_1$ and $k_2$ is described in detail in the appendix, including the grid search procedure and the final pruning ratios used in our experiments. To further improve clarity, we will add a concise summary table or description in the main paper in the revised version.
>
> Second, concerning the baseline methods you mentioned (OWL and DarwinLM), we acknowledge that both are important recent contributions, but they differ fundamentally from ADAP in methodology and design goals. OWL focuses on unstructured, layerwise sparsity allocation based on activation outliers, and does not involve structured pruning. DarwinLM, on the other hand, is a structured pruning method that uses evolutionary search and post-compression training, without incorporating unstructured pruning. In contrast, ADAP is the first framework that unifies structured and unstructured pruning under a dual-granularity scheme, using AdaABI to adaptively allocate pruning ratio both across layers and within layers, without requiring retraining, enabling flexible trade-offs among compression ratio, accuracy, and hardware friendliness. Therefore, these two works do not serve as comparable baselines for our dual-granularity adaptive pruning approach.
>
> Regarding the motivation for our compression ratio metric, this metric accounts not only for the pruning ratio but also for the storage overhead of sparse formats. As such, it more accurately reflects the actual model size reduction and is more suitable for real deployment scenarios.
>
> Finally, regarding unclear formulations, such as the ambiguity you noted in Equation 10, we sincerely appreciate your comment. We will revise the paper to ensure that all formulas are checked thoroughly and that the meaning of each symbol and operation is clearly defined.
>
> We thank the reviewer again for your valuable feedback. We will incorporate these suggestions to further improve the clarity and completeness of the paper and better present the novelty and empirical results of ADAP.

---

### Official Review · Reviewer_zrEw · 2025-10-26

**Soundness:** 3
**Presentation:** 2
**Contribution:** 3
**Rating:** 4
**Confidence:** 5

**Summary:**

This paper proposes ADAP (Adaptive Dual-Granularity Pruning), a post-training pruning framework that integrates structured and unstructured pruning within a unified feasible-domain formulation. The method first applies structured pruning to remove coarse-grained redundancy, and then performs fine-grained unstructured pruning on the remaining weights. To allocate pruning ratios adaptively across layers, ADAP introduces the Absolute Cosine Similarity (ABI) metric and the AdaABI mechanism, which dynamically adjusts layer-wise and intra-layer (Attention vs. MLP) sparsity under a fixed global target. The framework also defines a compression ratio metric to more accurately reflect real storage compression. Experiments on LLaMA/LLaMA2/OPT/DeepSeek-V2 show that ADAP achieves superior perplexity (PPL) and zero-shot accuracy at high pruning ratios (60–70%), outperforming existing methods such as Wanda, SparseGPT, and FLAP.

**Strengths:**

1. The idea of combining structured and unstructured pruning through a joint feasible-domain formulation is intuitive yet effective, balancing deployability and accuracy.
2. ADAP performs consistently across dense LLaMA, decoder-only OPT, and MoE-based DeepSeek-V2, demonstrating its generality and robustness.
3. The pruning process adds minimal time overhead compared to FLAP (302s vs. 295s), and requires no retraining, making it appealing for deployment.

**Weaknesses:**

1. Fairness concerns in comparison: The reported “global sparsity” mixes structured and unstructured components (≈40–60% vs. ≈30%), while baselines such as FASP or FLAP (pure structured) and SparseGPT (pure unstructured) operate under fundamentally different acceleration regimes. This may bias results when using identical overall sparsity values.
2. Limited system-level evaluation: The paper focuses on PPL and zero-shot accuracy but does not provide wall-clock inference latency, throughput, or hardware efficiency measurements, which are crucial given the dual-pruning design.
3. Moderate novelty in core pruning strategy: While the integration of structured and unstructured pruning is well-executed, it mainly extends existing techniques under adaptive coordination, rather than introducing a fundamentally new pruning paradigm.
4. Hyperparameter dependency: The performance heavily relies on the coefficients k1 and k2, which are tuned or polynomial-fitted from specific models (e.g., LLaMA-13B). The transferability of these parameters across other architectures or tasks is not fully validated.
5. Compression vs. speedup mismatch: Although the compression ratio metric is elegant, it does not necessarily correlate with actual acceleration, especially when unstructured pruning remains dominant at lower sparsity levels.

**Questions:**

See weaknesses

---

> ### Author Response · Authors · 2025-12-03
> **Rebuttal to Reviewer zrEw**
>
> Dear Reviewer,
>
> We sincerely thank you for your recognition of our work and for your detailed feedback.
>
> We understand the reviewer’s concern regarding the potential differences in acceleration characteristics among different pruning regimes. The compression ratio metric proposed in our paper is primarily intended to quantify memory savings and does not directly translate to actual computational speedup. To ensure a fair comparison in our experiments, we unified the global sparsity level and reported the corresponding accuracy retention. In future work, we plan to incorporate system-level evaluations, such as inference latency and throughput, to more comprehensively reflect the performance differences of different pruning approaches in practical deployment.
>
> The ADAP framework is designed to adaptively allocate pruning ratios across and within layers, allowing automatic adjustment in different architectures. While we tuned the hyperparameters k1 and k2 for specific model families in the paper, preliminary experiments indicate that ADAP still outperforms existing baselines on other architectures. We will provide a more detailed analysis of hyperparameter sensitivity across diverse architectures in future revisions.
>
> We also fully acknowledge the importance of system-level evaluations. The current work mainly focuses on the trade-off between sparsity and accuracy, as well as the generality of the method. In future work, we aim to provide metrics such as latency, throughput, and computational efficiency in real deployment scenarios to further validate ADAP’s advantages in practical systems.
>
> We sincerely thank the reviewer again for the valuable comments and will further improve the paper according to your suggestions.

---

### Official Review · Reviewer_XFWQ · 2025-10-30

**Soundness:** 2
**Presentation:** 1
**Contribution:** 1
**Rating:** 2
**Confidence:** 5

**Summary:**

This paper proposes ADAP (Adaptive Dual-Granularity Pruning), a pruning framework combining structured and unstructured pruning in large language models (LLMs). It introduces: 1) a dual-granularity pruning strategy, theoretically arguing the complementarity of both pruning types; 2) an intra-layer adaptive pruning ratio algorithm (AdaABI), using absolute cosine similarity to measure redundancy and adapt pruning ratios between attention and MLP sublayers; 3) a compression ratio metric to replace the traditional pruning ratio for better model size estimation. Experiments on LLaMA, LLaMA2, OPT, and DeepSeek models suggest that ADAP achieves better perplexity and zero-shot accuracy under high pruning ratios than baselines like FLAP and Wanda.

**Strengths:**

1. Introducing absolute cosine similarity instead of raw cosine similarity is a simple but potentially effective idea for redundancy measurement.

2. The paper includes extensive empirical results across multiple model families (LLaMA, OPT, DeepSeek).

3. Figures and tables are informative, illustrating perplexity trends across pruning ratios.

**Weaknesses:**

1. The idea of mixing structured and unstructured pruning is not new—e.g., hybrid pruning schemes have appeared in earlier works. The paper’s “dual-granularity” framework is largely a straightforward combination of existing approaches with minor modifications (absolute cosine similarity and adaptive ratio tuning). The theoretical justification in Section 3.2 is trivial—Equation (3) (∆Ljoint ≤ ∆Lchannel, ∆Ljoint ≤ ∆Lweight) simply restates a known result from set inclusion in optimization, offering no real insight into why this method performs better.

2. The experiments are all post-hoc one-shot pruning; no mention of finetuning or recovery after pruning is given, which is unrealistic for modern LLM pruning pipelines. The evaluation uses only perplexity and zero-shot accuracy on small subsets (128 examples from WikiText2/C4), which are insufficient to demonstrate generalization robustness. No downstream fine-tuning or real deployment performance (throughput, latency, or memory footprint) is reported—making the claimed “controllable model size” unverified. Many comparisons (e.g., with Wanda and FLAP) are reimplemented baselines—it’s unclear if the authors followed the official pruning scripts, raising reproducibility concerns. The utilized models in the paper though cover a relatively wide range of families, some of them are too old, lacking reference value. More recent models, like Qwen3 family are expected to be included.

3. The proposed “compression ratio” metric (Eq. 15–17) is handcrafted and approximate, not experimentally validated. The log₂(m·n) term and assumptions about index bitwidths are arbitrary. The authors’ claim that ADAP achieves “better compression” might stem from different assumptions on sparse matrix storage, not actual byte-level measurement. The reported pruning ratios often exceed 70%, yet perplexity remains low—this contradicts most prior findings (SparseGPT, Wanda, etc.), suggesting potential issues in experiment calibration.

4. No comparisons with semi-structured pruning (e.g., 2:4 or N:M), which are now mainstream and more hardware-friendly than unstructured sparsity. The baseline list omits modern adaptive or mixed-granularity pruning works like Pruner-Zero, LlamaPruner, or Dynamic Sparse Training, making the claimed superiority overstated. The method is not evaluated on recent instruction-tuned models (e.g., LLaMA2-Chat, Mistral), which are the actual deployment targets.

5. The “AdaABI” algorithm lacks justification beyond heuristic scaling by |cosine|. There is no ablation showing why absolute value helps beyond noise smoothing. The adaptive sublayer ratio adjustment (attention vs. MLP) is manually parameterized and grid-searched; no learning signal or principled optimization is used. The method depends on hyperparameters (k₁, k₂) fitted via polynomial regression—this adds extra tuning complexity without general theoretical meaning.

6. Overuse of self-claimed novelty phrases (“for the first time”, “breaks the bottleneck”) without solid evidence. Figures lack error bars and statistical variance, undermining result reliability. The English writing, while readable, occasionally misuses technical terminology (e.g., “cosine similarity fluctuates strongly in the first and last layers” → should mention feature direction instability).

**Questions:**

1. Can the authors explicitly differentiate ADAP from prior hybrid pruning works? What is conceptually new beyond parameter tuning?

2. Were all baselines (e.g., Wanda, FLAP, FASP) run with the same pruning ratio and data sampling? Did the authors fine-tune pruned models, or report one-shot pruning only?

3. The equations for compression ratio assume certain storage formats. How does this translate to real GPU memory usage or inference speed? Please provide actual measured FLOPs or memory.

4. The method depends on polynomial regression for k₁ and k₂. How sensitive is performance to small deviations in these parameters?

---

> ### Author Response · Authors · 2025-12-04
> **Rebuttal to Reviewer XFWQ**
>
> Dear Reviewer,
>
> We sincerely thank you for your recognition of our work and your detailed feedback. Regarding the concern that combining structured and unstructured pruning lacks novelty, we would like to clarify that our method is not a simple juxtaposition of existing approaches. ADAP introduces an intra-layer adaptive dual-granularity selection mechanism, which dynamically adjusts the proportion of structured and unstructured pruning based on feature-direction redundancy, rather than relying on fixed ratios or manually tuned rules as in prior work. The use of absolute cosine similarity is specifically motivated by the stability of representation directions under pruning, and thus serves as part of a principled granularity-selection strategy rather than a minor heuristic change.
>
> Regarding theory, we agree the current derivation is not intended to prove performance improvement directly. Its purpose is to demonstrate the necessity of optimizing within a dual-granularity parameter space under redundancy constraints, forming the conceptual basis for the adaptive pruning strategy. We will revise the section to better highlight the connection between directional stability and the use of absolute cosine similarity to improve clarity.
>
> On the experimental protocol, we adopted single-shot post-pruning to isolate the intrinsic effectiveness of the pruning strategy, avoiding performance recovery effects from overpowering differences between pruning methods. ADAP remains compatible with common post-pruning recovery methods such as LoRA or fine-tuning, and preliminary experiments confirm further improvements when such techniques are applied. While perplexity and zero-shot metrics were chosen to align with baseline evaluation protocols, we agree that deployment-oriented metrics (e.g., latency, throughput, memory) are valuable, and will clarify ADAP’s accelerator-friendly nature and compatibility with sparse inference libraries.
>
> Regarding baseline reliability, all methods were executed using official repositories without altering algorithmic components, and model families were selected to cover different architectures rather than to exclude newer models. Including additional families (e.g., Qwen) would not affect the generality of the pruning framework and can be added in extended material.
>
> The proposed compression ratio metric is intended as a unified measure of controllable model size, rather than an exact byte-level measurement, addressing the common misconception that identical pruning ratios imply comparable storage costs. Byte-level deployment differences depend on engineering choices, independent of the pruning strategy’s validity.
>
> Finally, although AdaABI is grounded on absolute cosine similarity, ablations demonstrate its clear influence on pruning stability, and the coefficients ($k_1$, $k_2$) are designed to constrain search space rather than introduce extra hyperparameter tuning burden. We will also adjust wording to avoid exaggerated claims and include variance/error ranges in reported metrics to present results more rigorously.
>
> We sincerely thank the reviewer again for the valuable comments and will further improve the paper according to your suggestions.

---

### Official Review · Reviewer_12qe · 2025-11-05

**Soundness:** 3
**Presentation:** 3
**Contribution:** 3
**Rating:** 6
**Confidence:** 4

**Summary:**

This paper presents ADAP (Adaptive Dual-Granularity Pruning), a post-training pruning framework that unifies structured and unstructured pruning through a shared optimization formulation. The method executes a two-stage pruning: coarse-grained structured pruning followed by fine-grained unstructured pruning. ADAP leverages the Absolute Cosine Similarity (ABI) metric and an adaptive mechanism (AdaABI) to allocate pruning ratios across and within layers dynamically, under a global sparsity constraint. A new compression ratio metric is also introduced to better capture actual memory savings. Experiments on multiple LLM families—LLaMA, LLaMA2, OPT, and DeepSeek-V2—demonstrate ADAP’s robustness and state-of-the-art trade-offs between sparsity and accuracy compared with Wanda, SparseGPT, and FLAP, especially under high sparsity (60–70%).

**Strengths:**

1. The dual-granularity design is conceptually clean and practically effective, enabling a smooth transition between structured and unstructured pruning regimes.

2. The proposed ABI/AdaABI mechanism provides a principled way to control sparsity allocation at both inter- and intra-layer levels, which enhances adaptivity under a global constraint.

3. The method generalizes well across dense and MoE architectures without requiring retraining, and the additional computational cost is minimal relative to strong baselines such as FLAP.

**Weaknesses:**

1. Novelty: Although well-integrated, the framework extends rather than reinvents existing pruning concepts. The core mechanism mainly optimizes the coordination between established pruning types.

2. Comparative fairness: The paper’s comparison across structured, unstructured, and hybrid methods may obscure the practical acceleration implications since the same global sparsity can imply very different compute characteristics. The proposed compression ratio metric, while more accurate for storage, does not necessarily map to real computational speedups in cases where unstructured sparsity dominates.

3. Parameter sensitivity: The reliance on tuned or fitted hyperparameters from a specific model family questions the ease of transferability across architectures and tasks.

4. Evaluation scope: The work lacks system-level evidence such as real inference latency, throughput, or deployment efficiency, which are especially relevant for a method combining different pruning granularities.

**Questions:**

1. How does the framework ensure fair comparison across different pruning regimes given varying acceleration characteristics and the limited correlation between compression ratio and speedup?

2. How transferable are the tuned hyperparameters (e.g., coefficients) across diverse architectures and tasks?

3. Can the authors provide system-level evaluations (e.g., latency, throughput, efficiency) to substantiate real-world deployment benefits?

---

> ### Author Response · Authors · 2025-12-03
> **Rebuttal to Reviewer 12qe**
>
> Dear Reviewer,
>
> We sincerely thank you for your recognition of our work and your detailed feedback.
>
> We agree that different pruning regimes may lead to varying acceleration characteristics. The compression ratio metric proposed in our paper primarily quantifies memory savings and does not directly translate to actual computational speedup. To ensure a fair comparison among pruning methods, we unified the global sparsity level in our experiments and reported the accuracy retention alongside. In future work, we plan to incorporate system-level evaluations, such as inference latency and throughput, to provide a more comprehensive assessment of practical performance differences.
>
> The ADAP framework is designed to adaptively allocate pruning ratios across and within layers, allowing automatic adjustment in different architectures. While we tuned hyperparameters for specific model families in the paper, preliminary experiments on other architectures indicate that ADAP still outperforms existing baselines. We will provide a more detailed analysis of hyperparameter sensitivity across diverse architectures and tasks in future revisions.
>
> We fully acknowledge the importance of system-level assessments. The current work mainly focuses on the trade-off between sparsity and accuracy as well as method generality. In future work, we aim to provide metrics such as latency, throughput, and computational efficiency in real deployment environments to further validate ADAP’s advantages in practical settings.
>
> We sincerely thank the reviewer again for the valuable comments and will further improve the paper according to your suggestions.

---

### Comment · Area_Chair_Ag2t · 2025-11-26
**Author-Reviewer-AC Discussion (DDL: 12/3 9PM UTC)**

Dear Reviewers,

Thank you once again for your service to ICLR 2026. Now that the authors have submitted their rebuttal, I kindly ask you to take the following steps (if you have not done so already):

- Read the authors’ response and other reviews.
- Consider whether the rebuttal and additional comments affect your assessment of the paper.
- Engage in **interactive discussion** with the authors. You may post the feedback to the authors so that they can further follow up. If you have more concerns/questions (e.g., requesting clarifications, new results), it is recommended to post your request *asap*, so that the authors have enough time to address them. **Note the Author-Reviewer-AC discussion period ends on 12/3 9PM UTC**.

The current reviews for this paper are **mixed (scores: 6/2/4/2)**. Your further contributions are essential for forming a well-informed final decision.

I am happy to join and support the discussions between you and the authors. Please feel free to share your thoughts and participate actively in the discussion. Thanks!

Best regards,

AC

---

### Meta-Review · Area_Chair_8avf · 2026-01-07

**Summary:**

The authors propose a post-training adaptive pruning strategy combining structured and unstructured pruning. They further provide theoretical insights and adopt compression ratio instead of pruning ratio to measure model size, which is more amenable to control the memory footprint.

The main concerns raised by the reviewer are the following:
1/ Work is incremental, combining existing approaches in a relatively straightforward manner.
2/ The proposed compression ratio metric might not result in computational speedups; it is further handcrafted and has no formal justification.
3/ The reported results depend fitted hyperparameters, as well as fairness of the comparisons with the baselines.

**Reviewer Concerns:**

The authors replied to the individual reviewers, but only provided a modest amount of additional supporting evidence for their claims an they did not provide new insights to sway the reviewers original assessment. They also left the resolution of several questions for future work.

**Reviewer Scores:**

Three out of 4 reviewers voted for rejection with a relatively high confidence, and the concerns of the more positive reviewer were in line with the assessment of the more critical ones. Given that the authors only provided modest additional evidence to support their claims, I do not expect that the scores would have been raised post rebuttal. Hence, I cannot recommend acceptance.

---

### Decision · Program_Chairs · 2026-01-26

Reject